# Human Endogenous Retrovirus as Therapeutic Targets in Neurologic Disease

**DOI:** 10.3390/ph14060495

**Published:** 2021-05-24

**Authors:** Karen Giménez-Orenga, Elisa Oltra

**Affiliations:** 1Escuela de Doctorado, Universidad Católica de Valencia San Vicente Mártir, 46001 Valencia, Spain; karen.gimenez@mail.ucv.es; 2School of Medicine and Health Sciences, Universidad Católica de Valencia San Vicente Mártir, 46001 Valencia, Spain; 3Centro de Investigación Traslacional San Alberto Magno, Universidad Católica de Valencia San Vicente Mártir, 46001 Valencia, Spain

**Keywords:** amyotrophic lateral sclerosis, epigenetics, HERV-K, HERV-W, monoclonal antibody, multiple sclerosis, neurodegeneration, temelimab

## Abstract

Human endogenous retroviruses (HERVs) are ancient retroviral DNA sequences established into germline. They contain regulatory elements and encoded proteins few of which may provide benefits to hosts when co-opted as cellular genes. Their tight regulation is mainly achieved by epigenetic mechanisms, which can be altered by environmental factors, e.g., viral infections, leading to HERV activation. The aberrant expression of HERVs associates with neurological diseases, such as multiple sclerosis (MS) or amyotrophic lateral sclerosis (ALS), inflammatory processes and neurodegeneration. This review summarizes the recent advances on the epigenetic mechanisms controlling HERV expression and the pathogenic effects triggered by HERV de-repression. This article ends by describing new, promising therapies, targeting HERV elements, one of which, temelimab, has completed phase II trials with encouraging results in treating MS. The information gathered here may turn helpful in the design of new strategies to unveil epigenetic failures behind HERV-triggered diseases, opening new possibilities for druggable targets and/or for extending the use of temelimab to treat other associated diseases.

## 1. Introduction

At least 45% of the human genome is comprised of transposable elements (TEs). They are classified into DNA transposons (sequences changing their positions by excising from one region to integrate into another) and retrotransposons (those needing to “make a copy” of themselves before being integrated). Retrotransposons can be categorized according to the presence/absence of long terminal repeats (LTRs) as LTR retrotransposons and non-LTR retrotransposons, respectively. Long Interspersed Nuclear Elements (LINE) and Short Interspersed Nuclear Elements (SINE) make up the non-LTR group, representing about 33% of the human genome, whereas human endogenous retroviruses (HERVs), flanked by LTRs, represent almost 8% [1,2,3].

HERVs are remnants of ancient exogenous viral infections that established the Vertebrates germline millions of years ago [4,5,6,7]. They classify into different families named after the amino acid one-letter code of the aminoacyl-tRNA used to prime their retrotranscription [2]. HERVs typically encompass three main proviral genes: *pol*, *gag* and *env,* flanked by two LTR sequences. The *gag* gene encodes structural proteins; the *pol* gene, a reverse transcriptase, a protease, and an integrase; and the *env* gene, an envelope protein. Since their introduction into the germline, some HERVs have accumulated mutations in their proviral genes, making them replication-defective because of frameshifts, the appearance of early stop codons or deletions/insertions of sequences. The homologous recombination between both LTR sequences has also led to a complete loss of proviral genes in some cases, leaving the HERV represented by its LTR only [3,4,5,7,8]. Through these mechanisms, evolution has left copies of TEs with variable genomic contents in the human population. An example is the newly discovered HERV-W copy on chromosome 18 that may be a homozygote solo LTR after complete proviral elimination, a heterozygote with one allele consisting in the remaining proviral genes and one solo LTR, and a homozygote with both alleles retaining the proviral genes [9]. Despite all these events, some families remain transcriptionally active, most being expressed in a tissue-specific manner [10,11,12], but also contribute to human genome heterogeneity [9,13].

HERVs have occasionally provided their hosts with beneficial effects. They contribute to genomic variability, provide additional gene regulatory elements, such as alternative promoters or alternative splice sites [14,15,16,17], or rule important physiological processes. HERVs also participate in transcriptional regulation with noncoding RNAs [18,19] or in the expression of “domesticated” viral proteins. For example, an element of the HERV-W family, ERVWE1, encodes Syncytin-1, a protein essential for embryo implantation [20,21]. Additionally, some HERVs are proposed to protect the fetus by suppressing maternal immunological reactions [22,23]. Nonetheless, if uncontrolled and/or abnormally activated, HERVs can lead to detrimental effects in their host organisms by altering gene expression profiles, expressing pathogenic nucleic acids/proteins or even inducing deleterious mutations [2,24,25] (Table 1).

## 2. Epigenetic Control of HERVs

The expression of HERVs can be controlled at different levels, but the most important are the epigenetic regulatory mechanisms [47,48] (Figure 1). While most HERVs are constitutively repressed in the human genome, some co-opted elements must become activated in a limited way, at specific time frames and in particular cell types [20,49]. To achieve this complex context-specific regulation, some HERVs harbor binding sequences for KRAB-containing zinc finger proteins (KRAB-ZFPs). Interestingly, the number of KRAB-ZFPs correlates with the LTR transposon load across species [50]. In fact, these transcriptional repressors can benefit the host via the domestication of TEs, including HERVs [51].

KRAB-ZFPs are comprised of an array of C2H2 zinc fingers, which confers them with specificity to bind the DNA sequences lying within HERV elements, and a Krüppel-associated box (KRAB) that recruits the corepressor KRAB-associated protein 1 (KAP1), also known as TRIM28 [52,53,54,55,56]. KAP1 serves as a scaffold for the assembly of heterochromatin epigenetic modifiers like the histone methyl transferase SETDB1 (also known as ESET) [57], the heterochromatin protein 1 (HP1) [58] or DNA methyl transferases (DNMTS) [59,60] (Figure 1, number 1). Three molecules of KAP1 bind to the KRAB domain of the KRAB-ZFPs through their N-terminal RING-B box-coiled coil (RBCC) region, while the C-terminal PDH domain of KAP1 functions as an intramolecular E3 ligase that sumoylates several lysines in its bromodomain. Sumoylation is a post-translational modification that consists of the covalent attachment of a Small Ubiquitin-like Modifier (SUMO) to a lysine residue. KAP1 bromodomain sumoylation enhances its corepressor activity and enables its interaction with SETDB1 [54,61]. Furthermore, it enhances SETDB1 activity [62] (Figure 1, number 1). When SETDB1 interacts with KAP1, it methylates histone H3 at lysine 9 (H3K9me) through its C-terminal ubiquitinated SET domain [57,62]. This mechanism leads to the repression of a subset of HERVs where the H3K9me heterochromatin has not yet been established, as shown by the activation of a larger group of HERVs in SETDB1 knockouts as compared to KAP1 knockouts [48]. Furthermore, KAP1-mediated silencing is especially important for brain development. Its depletion in neural progenitor cells aberrantly activates ERV expression while having no impact on adult neural cells [63].

Also, SETDB1 can directly bind, in a KAP1-independent way, to H3K9me1/K14ac or H3K9me2/K14ac histone tails through its N-terminal triple Tudor domain (TTD) to deacetylate K14 via SETDB1-associated histone deacetylases (HDACs) and further methylate K9 to achieve a trimethylated (H3K9me3) state [64] (Figure 1, number 2). SETDB1 takes part in silencing retroelements in differentiated cells, including B cells [65] and brain tissue [66], as well as innate immune genes [56].

The introduction of H3K9me3 by SETDB1 also allows the binding of HP1 to the genome, where it interacts with KAP1 as well [52,57] (Figure 1, number 1). HP1 proteins prevent transcription factor binding and help maintain the heterochromatic structure [67]. Indeed, the HP1 isoform α may be required to repress the expression of both immune-related genes and HERVs [68]. Besides SETDB1 and HP1, KAP1 also recruits other repressor proteins, such as the NuRD HDAC complex, which removes transcription-promoting histone acetylation [69] (Figure 1, number 1). Moreover, since the axis KRAB/KAP1/SETDB1 localizes to restricted regions within HERVs, and H3K9me3 marks must spread out all over HERV sequences to ensure their silencing, KAP1 may form a complex with HP1 and histone methyltransferases all over the HERV region to mediate the spreading of heterochromatin [70,71].

Although some studies have highlighted the importance of histone post-translational modifications (PTMs) in regulating the HERV expression in terminally differentiated cells [53,60,65], DNA methylation seems to also play a relevant role in achieving their regulation [47,72,73,74]. Furthermore, the impact of DNA methylation-dependent mechanisms on HERVs expression may vary according to cell type and the evolutionary age of the element. For example, histone PTMs mediated by SETDB1, and not DNA methylation, are essential for HERV repression in neural progenitor cells [66,75] and B cells [65]. Additionally, younger LTRs are found to be suppressed by DNA methylation, while older ones are mainly silenced by histone PTMs [76]. By contrast, both mechanisms seem to play important roles in embryonic stem cells [77].

**Figure 1 pharmaceuticals-14-00495-f001:**
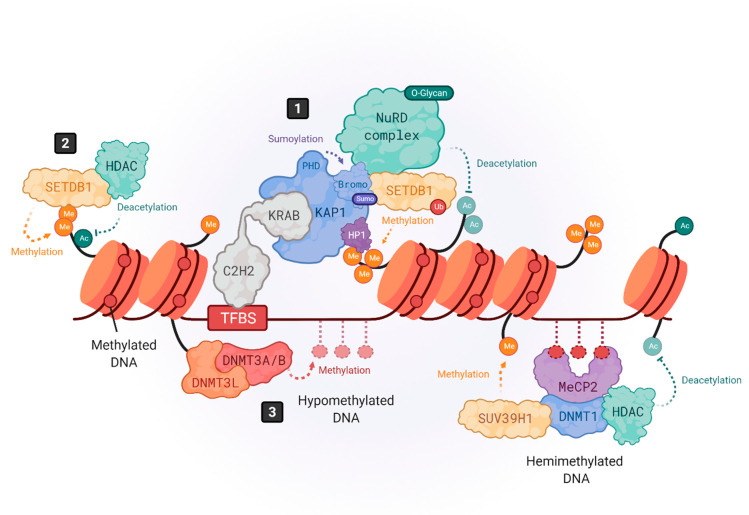
Epigenetic silencing mechanisms of HERVs. The expression of HERVs is mainly controlled by DNA methylation and histone tail modifications. Histone modifications are driven by KRAB-containing zinc finger proteins (KRAB-ZFPs), which bind to transcription factor binding site (TFBS) within HERV elements, recruiting the corepressor KRAB-associated protein 1 (KAP1), also known as TRIM28 (**1**) [52,54]. KAP1, in turn, serves as a scaffold molecule for the assembly of heterochromatin modifiers like the histone methyltransferase SETDB1 (also known as ESET) [57], the heterochromatin protein 1 (HP1) [58] or the NuRD histone deacetylase complex [69]. Moreover, SETDB1 can also bind in a KAP1-independent way to H3K9me/K14ac marks to drive histone methylation (**2**) [64]. DNA methylation is driven by DNA methyltransferases (DNMTs). DNMT3L binds to histone 3 tails and recruits DNMT3A and DNMT3B to establish de novo methylation in hypomethylated DNA (**3**), whereas hemimethylated DNA is bound by methyl-binding proteins like MeCP2, which recruits repressor complexes such as DNMT1, histone deacetylases (HDACs) and methyltransferases like SUV39H1 to stabilize the chromatin structure repressing the gene expression (**4**) [47,72,73,74]. Created with Biorender.com (accessed on 21 May 2021).

DNA methylation is a modification carried out by DNA methyltransferases (DNMTs) that consists on the transfer of a methyl group onto carbon 5 (C5) of cytosines to form 5-methylcytosine, thus recruiting methyl-CpG-binding proteins (MBP) and/or preventing the binding of transcription factors [72]. It has been shown that the brain contains some of the highest DNA methylation levels in the body and that MBPs are important for normal neuronal functions, being highly and specifically expressed in the brain [78]. DNA methylation can be promoted at specific regions of the genome through transcription factor or RNA interference (RNAi)-mediated mechanisms [79] or simply spread across susceptible genomic regions. In hypomethylated DNA, DNMT3L binds to histone 3 tails and recruits DNMT3A and DNMT3B to establish de novo methylation [72,80,81] (Figure 1, number 3).

Additionally, DNMT3A/B can bind hypomethylated DNA regions, unprotected by transcription factors. When DNA is hemimethylated, MBPs like MeCP2 bind to DNA and recruit various repressor complexes such as DNMT1, or H3K9 methyltransferases like SUV39H1, to achieve DNA and histone methylation [72,78,82,83,84,85], and histone deacetylases (HDAC) to remove activating histone modifications [86] (Figure 1, number 4). Thus, DNA methylation and histone modifications work closely to induce a heterochromatic-repressive state. In fact, one study demonstrated that DNMT3L facilitates the binding of KAP1 to various repressive epigenetic modifiers [87].

In addition to DNA methylation and histone modifications, transcription factor PTMs also play an important role in HERV silencing. As mentioned earlier, KAP1 sumoylation enhances its recruitment to retroelements but, also, the recruitment of SETDB1 to mediate H3K9 trimethylation [61] (Figure 1, number 1). In addition, the inhibitory activity of KAP1-associated repressive chromatin factors, such as the HDAC1 of the NuRD complex [88,89], is enhanced by methylation-dependent O-linked glycosylation, a PTM introduced by the O-linked β-*N*-acetylglucosamine transferase (OGT) [90] (Figure 1, number 1). On the other hand, HERVs can also be post-transcriptionally regulated by RNA-binding proteins, such as the RNA-Binding Motif Protein 4 (RBM4) [91]. RBM4 regulates HERV-K (HML-2) and the HERV-H transcript levels by binding at CGG consensus elements present on 5′ and 3′ LTRs. However, the mechanism by which RBM4 mediates HERV downregulation remains unknown. Another post-transcriptional regulatory mechanism is m6A RNA methylation, which reduces the half-life of ERV mRNAs, protecting cell integrity, as shown in mouse cells [92]. Thus, a combination of multiple heterochromatin modifiers, their PTMs, and the post-transcriptional regulation of HERV transcripts, may prevent the ectopic and/or pathogenic expression of their encoded nucleic acid sequences and proteins that could be sensed by the immune system, triggering inflammatory responses [24,25].

## 3. Transactivation of HERVs

Despite their tightly controlled regulation, pathogenic HERV proteins have been detected in chronic inflammatory or degenerative diseases of the nervous system, such as multiple sclerosis (MS) [93,94], amyotrophic lateral sclerosis (ALS) [41] or schizophrenia [95,96,97]. The main families being related to these neurological diseases are the HERV-Ws and the HERV-Ks [98].

The HERV-W family is present in the human genome in multiple copies, most of them being defective and replication-incompetent [99,100], even though LINE-1 elements could, in principle, ease their transposition [100,101]. Although some complete proviral genes have been detected in these elements [94,102] they do not seem to encode complete proteins, mainly due to the presence of early stop codons [103,104]. The only known HERV-W family member encoding a complete *env* gene that transcribes and translates into a complete Env protein is ERVWE-1, a replication-incompetent HERV-W element located on chromosome 7q21.2 [20,21,27] that encodes Syncytin-1, a protein found only intracellularly or on the plasma membrane and is, therefore, cell-associated. Syncytin-1 plays an important role during early pregnancy by facilitating trophoblastic cell fusion to allow embryo implantation [20,105]. It consists of a 538-aa polypeptide (GenBank accession no. AF072506) comprising a cell surface domain (SU) and a transmembrane domain (TM). The SU domain is known to interact with the sodium-dependent neutral amino acid transporters ASCT-1 and ASCT-2 expressed by trophoblasts [106], while the TM domain promotes cell–cell fusion to mediate syncytium formation [107].

By contrast, MS-associated retrovirus (MSRV), now referred to as HERV-W/MSRV, have been found extracellularly, in the cerebrospinal fluid (CSF) of MS patients, as genomic RNA assembled into proviral particles [28,93,108]. The origin of HERV-W/MSRV remains uncertain, since any known HERV-W family member could, in principle, yield viral particles, and no human genome locus encoding HERV-W/MSRV has yet been found in the human genome. Thus, it has been suggested that it originates from the complementation of different HERV-W elements, by reversing the stop codon in the Xq22.3 locus [104,109], from a non-ubiquitous and/or unfixed copy [9,110] or could even be an exogenous member of the HERV-W family [108,111].

The HERV-W/MSRV envelope protein (GenBank accession no. AF331500), named pHERV-W ENV from the pathogenic HERV-W envelope protein in MS [112], consists of a 542-aa polypeptide with six potential N-glycosylation sites, which is divided by a cleavage site into a SU domain of 290 aa and a TM domain of 252 aa [108]. Interestingly, and likewise for all HERV-W sequences, the pHERV-W ENV length only differs from Syncytin-1 by the presence of four additional amino acids. Despite a reported identity of 81% [21], several important differences between these two proteins have been detected. Firstly, while the Syncytin-1 TM domain has fusogenic properties, the pHERV-W ENV fusogenic region is nonfunctional. Secondly, Syncytin-1 is a membrane-associated protein, while pHERV-W ENV can be membrane-associated within viral particles or as a hexameric-soluble form, although monomer and trimers have also been described [113,114]. In the hexameric-soluble form, the hydrophobic TM domain of each monomer is located toward the inside, at the core of the complex, whereas the hydrophilic SU domain is exposed to the surrounding biological fluids [113,114]. Thirdly, HERV-W/MSRV is the main family member associated with chronic inflammatory diseases, though ectopic pathogenic Syncytin-1 expression has been reported in schizophrenia [115].

Regarding the HERV-K family, the youngest in the human genome [6], it is composed by ten groups (HML-1–HML-10), although an HML-11 group has more recently been identified [116]. The HML-2 group comprises the most recent set of integrations in humans, with more than 90 proviruses that maintain ORFs (open reading frames) encoding the functional proteins [116]. HERV-K is the only known family able to produce retroviral particles; however, the particles produced do not seem to be infectious [117,118,119,120]. Although it is not clear which genomic sequences are responsible for viral particle production, it is known that HERV-K (HML-2) is abnormally expressed in the neurons of ALS patients [121].

The activation of the HERV-W and HERV-K sequences may derive from dysregulation of the epigenetic mechanisms controlling their expression, since the inhibition of DNMTs and HDACs leads to HERV reactivation [122]. As epigenetic mechanisms can be influenced by environmental stimuli, it seems possible that some environmental factors may distort HERV silencing mechanisms, reactivating their expression and, thereby, inducing or contributing to disease. In fact, some viral infections have been shown to lead to reactivation of HERVs and, interestingly, are believed to be cofactors of HERV-associated diseases. For example, *Herpesviridae* infections are associated with a higher risk for developing MS [94,123,124,125], while influenza seems to contribute to schizophrenia, especially when infections occur during the prenatal stages [126,127,128,129].

Influenza virus has been shown to transactivate HERV-W elements by impairing the KRAB-ZFP/KAP1/SETDB1 axis, which plays an important role in controlling HERV expression, as described above (Figure 1, number 1). In vitro, influenza virus triggers loss of sumoylations on KAP1 and lowers the levels of H3K9 trimethylation and SETDB1 without affecting promoter DNA methylation status. This leads to de-repression of HERV-W elements like Syncytin-1 [130,131,132].

Besides, herpes viruses like Herpes Simplex Virus type 1 (HSV-1), Human Herpes Virus type 6 (HHV6) and Epstein-Barr Virus (EBV) are efficient at activating HERV-W elements, even on methylated promoters [113,133]. In fact, in vitro infection by HSV-1 was shown to transactivate HERV-W by stimulating LTR-directed transcription [134], inducing its expression in neuronal and brain endothelial cells [135]. Another HERV-W activator is HHV6 [27,136]. For the Epstein–Barr virus (EBV), in vitro experiments showed that the binding of the EBV major envelope protein gp350 activates the transcription of HERV-W/MSRV and Syncytin-1 in peripheral blood mononuclear cells (PBMCs) and astrocytes without requiring EBV entry [94]. Indeed, in vivo infection triggered higher MSRV activation in PBMCs when patients had or had passed symptomatic infectious mononucleosis, the clinical manifestation of EBV [137].

EBV and HHV6 not only transactivate HERV-W elements but, also, HERV-Ks [138,139,140,141,142]. Further, first-episode schizophrenia is associated with a reduction of HERV-K methylation in PBMCs [143]. However, not only viral infections have the capacity for HERV activation; cytokines can also induce their expression. HERV-K LTRs appear to be responsive to proinflammatory stimuli in ALS [144]. This opens up the intriguing possibility that the activation of HERVs in inflammatory diseases serves to create a feedback loop amplifying abnormal immune responses when pathogenic HERVs have been epigenetically de-repressed and activated. This may be readily observed when HERVs are not constitutively repressed, as it occurs with certain HERV-K copies that are accessible to cytokine-triggered signaling pathways [145] or with HERV-W elements in epigenetically primed EBV-transformed B cells [111].

Like MS, no single environmental stimuli or single gene mutations have been described as the disease-causing agents for schizophrenia or ALS, but a combination of environmental factors and genetic susceptibility backgrounds have been proposed [146,147]. It is tempting to think that HERV activation may represent the common denominator to those risk factors, with particular combinations leading to the reactivation of specific HERV elements. In this hypothetical scenario, depending on the HERV(s) element(s) de-repressed and the cell type(s) involved, the resulting symptoms and, thereby, the disease triggered would be one or another.

## 4. HERV-Associated Pathogenesis

### 4.1. HERV-W Related Diseases

Both HERV-W envelope proteins, pHERV-W ENV and Syncytin-1, have been shown to possess immunopathogenic features in vitro and in vivo [24,29,93,148,149]. One of the most studied pathologies linked to abnormal HERV expression is MS, an autoimmune neurodegenerative disease that affects myelin sheaths, leading to neuronal loss [93,150]. The commonly used experimental model for MS is the mouse autoimmune model EAE (Experimental Autoimmune Encephalomyelitis) [151], from which most of the knowledge about neuroinflammation and neurodegeneration derives. EAE is induced by immunization with myelin or central nervous system (CNS)-derived peptides, and the disease is mediated by activated CNS-specific CD4+ T cells in the periphery, which enter the CNS by crossing the endothelial blood–brain barrier (BBB). By contrast, neuroinflammation in MS has now been revealed to be initiated in the CNS with the activation of microglia [152,153]. Under physiological conditions, microglia protect the brain by sensing pathogens, phagocytosing cell debris and promoting remyelination. However, under MS pathological conditions, microglia become activated and secrete proinflammatory molecules that directly damage myelin sheaths and oligodendrocytes, causing inflammation and neurodegeneration [154,155] (Figure 2, number 1). This proinflammatory environment attracts peripheral monocyte-derived macrophages and dendritic cells (DC), as well as peripheral autoreactive T-lymphocytes, which enter the CNS through the BBB localized in CNS microvessels [152,156,157,158] (Figure 2, number 2). Infiltrating T-lymphocytes become activated by microglia, which also act as antigen-presenting cells (APCs), phagocytosing myelin debris and expressing major histocompatibility complexes (MHC) I and II with costimulatory molecules (Figure 2, number 3). Infiltrated monocyte-derived DCs also act as APCs, presenting myelin antigen to T cells, along with microglia [159] (Figure 2, number 3). Activated T-lymphocytes differentiate into T-helper cell-type 1 (Th1) (producing IFN-γ) and Th17 cells (producing IL-17, IL-22 and IL-21) [152,154,155,160,161] (Figure 2, number 3). The crosstalk between neuroimmune and peripheral immune systems exacerbates neuroinflammation, further activating microglia and astrocytes through cytokine production, as well as facilitating the infiltration of other immune cells, such as activated B cells, monocytes and macrophages. The release of proinflammatory cytokines, nitric oxide, reactive oxygen and nitrogen species [162], together with some other processes, like phagocytosis of myelin by macrophages and antibody-dependent cytotoxicity, contribute to inflammation within the CNS, oligodendrocyte loss, and demyelination of the white and gray matter [163].

Although neurodegeneration has traditionally been thought to be a consequence of inflammation in the CNS, this concept has been challenged by the poor efficacy of immunomodulatory treatments on preventing demyelination and axonal degeneration [164]. Now, we know that HERV-W/MSRV, through the expression of pHERV-W ENV, may participate in the development of MS by triggering a potent immune response, as well as by impairing the myelin-repairing process [24,25,112,164,166,167,168], which could explain why immunomodulatory treatments failed at preventing neurodegeneration. Furthermore, pHERV-W ENV contribution to MS is supported by the observation that administered pHERV-W ENV with myelin oligodendrocyte glycoprotein (MOG) 35–55 peptide triggers EAE in mice [168].

pHERV-W ENV RNA and protein have been detected in PBMCs of MS patients [27,28,111,169], showing their highest RNA levels in natural killer (NK) cells, followed by B cells and monocytes, the last increasing their expression by differentiation to macrophages. By contrast, no ENV-encoding RNA or protein is detected in T cells [29,94]. pHERV-W ENV has also been detected in astrocytes [94,105], infiltrated macrophages and activated microglia in the brains [165,169,170] of MS patients, and its soluble form in their serum, plasma, and CSF [28,93,171,172,173,174]. Indeed, the presence of HERV-W RNA in CSF is considered as a negative prognostic marker of MS. Its load increases with MS duration and parallels the clinical stages [172,173,174,175,176].

The SU domain of the pHERV-W ENV, either associated to viral particles or in its soluble form has been shown to trigger proinflammatory responses in vitro, in human PBMCs and dendritic cells (DCs) [25] and, in vivo, in humanized severe combined immunodeficiency (SCID) mice, which die from brain hemorrhages with an overexpression of TNF-α after pHERVW-ENV injection [24,149]. The inflammatory reaction induced by pHERV-W ENV is mediated by the activation of Toll-like receptor (TLR) 4 and its coreceptor CD14 [25,167]. TLRs are expressed by different cell types, such as B cells [177], macrophages, monocytes, DCs [178], oligodendrocytes [179], astrocytes [180] or microglia [181], among other, where they mediate part of the innate immune response against invading pathogens [182,183].

On the one hand, pHERV-W ENV activates monocytes, DCs and macrophages (Figure 2, number 4–6). Activated monocytes produce proinflammatory cytokines, including TNFα, IL-1β, IL-6 and IL-12p40 (Figure 2, number 4). Interestingly, the levels of IL-6 and IL-12p40 induced in cultured PBMCs directly correlate with the MS disease severity [25]. Activated DCs not only release the proinflammatory cytokines IL-6, TNFα, IL-12p40 and IL-12p70 but, also, upregulate antigen presentation, activating antigen-specific T cells. The release of IL-12 by DCs activates naïve T cells and promotes their differentiation into IFN γ-secreting Th1 cells [25] (Figure 2, number 3). In turn, released INF-γ and TNF-α upregulate pHERV-W ENV secretion by PBMCs [111]. Thus, pHERV-W ENV not only activates the innate immune system but, also, the adaptative immune system. Moreover, it also acts like a superantigen [24], stimulating abnormal polyclonal T-cell activation through its binding to Vβ chains, and the expansion of T-cell-receptor β-chain (TCR Vβ) cells, leading to nonspecific oligoclonal Vβ+T cell activation [24,184] (Figure 2, number 7).

In addition, pHERV-W ENV activates vascular endothelial cells within the BBB, leading to the release of proinflammatory cytokines IL-6 and IL-8 and to the overexpression of ICAM-1, thus enhancing the migration of circulating cells towards the brain (Figure 2, number 8) [164]. In the brain, the pHERV-W ENV-SU domain triggers CNS innate immunity through TLR4 binding, activating microglia and perivascular macrophages, which release proinflammatory cytokines, leading to neuroinflammation and neurodegeneration [185] (Figure 2, number 1). Additionally, pHERV-W ENV produced by the surrounding cells like microglia or macrophages [27] engages with, and activates, TLR4 receptors present on the brain’s resident oligodendrocyte precursor cells (OPCs). The OPCs’ role is to migrate to the lesion sites where oligodendrocytes are depleted and differentiate into mature oligodendrocytes to remyelinate axons after an injury [186]. After migration, OPCs transiently express TLR4 receptors, which become downregulated during the OPCs’ differentiation process to oligodendrocytes [165]. TLR4 activation does not affect OPC survival; however, it induces the release of proinflammatory cytokines such as TNF-α, IL-1β and IL-6, as well as the overexpression of inducible nitric oxide synthase (iNOS), leading to an increase in nitric oxide (NO) levels that triggers nitrosative stress, which, in turn, affects the myelin protein expression and reduces OPCs’ differentiation capacity [165] (Figure 2, number 9). NO and TNF-α are well-established mediators of axonal injury and demyelination [187,188]. Interestingly, TLR4 activation in OPCs seems to be sensitive to the cyclin-dependent kinase inhibitor p57kip2, which participates in controlling TLR4’s surface expression [166].

On the other hand, pHERV-W ENV induces a degenerative phenotype in microglial cells characterized by the expression of the same set of proinflammatory cytokines detected on pHERV-W ENV-activated OPCs (TNF-α, IL-1β and IL-6) and by the overexpression of iNOS [112] (Figure 2, number 1). While reducing anti-inflammatory and neuroprotective proteins and downregulating the expression of genes involved in myelin debris phagocytosis, it affects neurorepair and inhibits OPC differentiation. Furthermore, the pHERV-W ENV protein stimulates microglia proliferation and drives physical interaction between microglial cells and myelinated axons to induce leakage of intra-axonal and myelin proteins [112] (Figure 2, number 10). Of note, the multifaceted effects of pHERV-W ENV are often due to its ability to strongly bind TLR4 receptors on expressing cells, thereby activating underlying signaling pathways and leading to various effects with a proinflammatory common point in different cells [189].

Furthermore, pHERV-W-ENV inflammatory and demyelinating consequences can also be observed in chronic inflammatory demyelinating polyradiculoneuropathy (CIDP), a rare immune disease of the peripheral nervous system (PNS) [168]. In these patients, pHERV-W ENV is found in PBMCs and peripheral nerve lesions, especially in Human Schwann Cells (HSCs), in which it exerts pathogenic effects by TLR4 engagement, inducing the release of proinflammatory IL-6 and the macrophage and T-cell chemoattractant CXCL10 [190], triggering activation of the innate immune system.

Another HERV-W-related disease is schizophrenia [191,192], a psychiatric disorder that highly impacts patients’ quality of life. In reference to the mechanisms underlying this disease, several are the hypotheses raised, including the disturbance of the dopaminergic neurotransmission or its relationship with neurodevelopmental problems. However, the idea that autoimmune processes within the nervous system could be playing an important role in its pathophysiology has been gaining more and more acceptance [193]. Some studies have shown the elevated expression of TLRs [194] and abnormal expression of proinflammatory cytokines, like IL-6, IL-8 and the C-reactive protein (CRP), in the serum of these patients [195,196]. CRP is released to blood under inflammatory conditions, and its overexpression supports that immune mechanisms play a role in schizophrenia symptoms beyond cognitive decline [197]. Interestingly, the levels of HERV-W mRNA were found to associate with a proinflammatory phenotype in these patients [198].

HERV-W expression is particularly associated with recent-onset schizophrenia [31,32,95,191]. Both HERV-W/MSRV and Syncytin-1 were reported to be increased in schizophrenia [33,115], and HERV-W/MSRV was shown to be unequally represented in the genome of an affected and its nonaffected monozygotic twin [199]. HERV-W GAG and ENV RNA or protein have been found in the circulating blood of these patients [31,32,33,192,200], correlating GAG protein levels with disease severity [200], and Syncytin-1 or (MSRV) pHERV-W ENV with CRP levels [32], providing a link between HERV-W envelopes and the status of systemic inflammation in these patients [32,115,192]. Activation of CRP in microglia and astrocytes, and the consequent inflammation, can be triggered by Syncytin-1 through direct interaction with TLR3 receptors [115] and IL-6 release. Furthermore, HERV-W presence has also been detected in CSF [95] and in cortical tissue [201], indicating a pathogenic role in the brain. In fact, HERV-W/MSRV induces activation of the Brain-derived neurotrophic factor (BDNF) and Dopamine receptor D3 (DRD3) genes [33], both implicated in providing a higher risk for suffering schizophrenia [202]. Thus, HERV-W Syncytin-1 and MSRV may determine and/or promote the development of schizophrenia. In this regard, it should be mentioned that, by combining single-molecule tracking, calcium imaging and behavioral approaches, Johansson et al. [97] showed that the mechanism used by HERV-W-ENV to alter glutamate synapse maturation and generate behavioral deficits depends on the N-methyl-d-aspartate receptor (NMDAR) organization and cytokine-dependent changes, further supporting an etiological role for this HERV product in the development of psychosis.

### 4.2. HERV-K Related Diseases

The main disease related to HERV-K activation is ALS. It is a progressive motor neurodegenerative disease leading to cortical and spinal neurons degeneration, causing muscle atrophy, weakness and, ultimately, death [203]. About 10% of ALS cases are familial (fALS), and out of them, 20% are associated with mutation of the superoxide dismutase (SOD1) gene. However, the remaining ALS cases, which comprise the majority, are considered sporadic (sALS) because no genetic component has been identified [204]. Instead, sALS are related to the presence of pathological DNA-binding protein-43 (TDP-43) neuronal cytoplasmic aggregates, which are not present in SOD1-mutant fALS, suggesting different disease mechanisms [205,206].

TDP-43 was initially identified as a protein able to bind TAR DNA sequence motifs in the human immunodeficiency virus (HIV) genome [207]. Later, it was found to locate in the nucleus of neurons and glial cells, such as oligodendrocytes [206,208]; however, its physiological role in the brain remains unknown. Besides HIV, another retroviral element containing TDP-43-binding sites is the endogenous HERV-K (HML-2). Interestingly, HERV-K (HML-2) RNA is upregulated in PBMCs of HIV patients [209]. TDP-43 levels in the cortical and spinal neurons of ALS patients correlate and colocalize with HERV-K expression [41,121]. Although TDP-43 has been found to bind HERV-K LTRs, only the ALS-associated pathological TDP-43 aggregates seem to have the potential to induce HERV-K transcription in cortical neurons [41,210]. The transcripts of HERV-K *GAG, POL* and *ENV* genes have been detected in neuronal cells of postmortem ALS brains [41], and, particularly, the expression of the HERV-K Env protein triggers neurite retraction in in vitro neuronal cultures. Furthermore, its overexpression in vivo decreases the number of cortical and spinal neurons of transgenic mice, which also presents with shorter dendrites, less spines, and a lower complexity, leading to motor dysfunction and, eventually, death [41]. The mechanisms by which HERV-K Env protein exerts those neurotoxic effects are unknown. However, it has been suggested that it could be impairing the normal function of the nucleolar protein nucleophosmin leading to nucleolar dysfunction, a feature observed in ALS patients with mutations on the ORF 72 gene of chromosome 9 (*C9ORF72*), the most frequent genetic defect in ALS [41,211]. Additionally, HERV-K (HML-2) Env downregulation is necessary for the differentiation of neural stem cells [212]. In this regard, transcriptional activation of the HERV-K (HML-2) group of elements impairs cortical neuronal differentiation, reducing neurites length and complexity. This effect was reported to be mediated by upregulation of the Neurotrophic Receptor Tyrosine Kinase 3 (*NTRK3*) gene [213], an important factor for the development of cortical neurons whose knockdown completely rescued this phenotype [213,214].

HERV-K (HML-2) is also involved in Alzheimer’s disease, an illness characterized by the accumulation of beta-amyloid protein aggregates (plaques) in the extracellular matrix and Tau protein aggregates (tangles) inside neurons. The progressive accumulation of these proteins causes synaptic loss and neurodegeneration [215]. HERV-K (HML-2) transcripts are found upregulated in the CSF of Alzheimer’s patients in correlation with TLR8 RNA. HERV-K-induced neurotoxicity and neuronal apoptosis are mediated by TLR8 and SARM1 (Sterile alpha and TIR domain-containing 1) signaling. Furthermore, HERV-K RNA stimulates the release of proinflammatory molecules in microglia. However, microglia activation does not seem to be required for triggering neurotoxicity [216].

## 5. HERV-Targeted Treatments

With the discovery of the implication of HERVs in chronic diseases such as MS, schizophrenia or ALS, a new avenue for the development of targeted therapies directed against HERV proteins opened. In this direction, the Swiss company GeNeuro developed the first HERV-directed treatment, named GNbAC1 or temelimab, that consists of a humanized immunoglobulin (Ig) G4/ kappa (κ) isotype monoclonal antibody (mAb), with an approximate molecular weight of 147 kDa, directed against the pHERV-W ENV. Like other antibodies, temelimab is comprised of two heavy chains (HC), from class IgG subtype 4, bound to each other by disulfide bonds, with a light chain (LC)-type κ bound to each HC [217]. However, site-directed mutagenesis was performed to stabilize the interchain disulfide bridges. At the same time, each HC and LC are constituted by a constant domain that determines the biological properties of the antibody and a variable domain, which determines the binding specificity. While the LC includes a variable domain (VL) and a constant domain (CL), the HC includes a variable domain (VH) and three constant domains (CH1, CH2 and CH3). Moreover, at the N-terminal of the VL and VH lies the complementarity determining regions (CDRs), which define the structural complementarity between the antibody and the specific antigen that binds. The LC and HC each have three CDRs shaping the antigen binding site. Of note, temelimab is a humanized mAb because the variable and constant regions come from a human antibody, but only the CDRs derived from the parental murine antibody remain, which specifically bind the pHERV-W ENV.

Temelimab targets a linear non-glycosylated epitope of the SU domain of the HERV-W/MSRV Env [218,219,220], blocking its interaction with the TLR4 receptor and, hence, the proinflammatory cascade and the inhibition of the myelin repair process. Its testing in vitro proved that temelimab could prevent pHERV-W ENV-mediated PBMC activation, and the consequent TNF-α release, without blocking TLR4 or inhibiting its activation by other ligands, like lipopolysaccharides (LPS) [25]. In addition, this antibody also prevents the pHERV-W ENV-mediated release of proinflammatory cytokines, especially of TNF-α, by OPCs and the induction of nitrosative stress, neutralizing the pHERV-W ENV-associated inhibition of OPC differentiation [112,166]. Furthermore, when tested in vivo in the EAE mouse model [168], all the temelimab-treated mice healed and survived; in fact, even remyelination of the CNS lesions was observed [219,220]. These results further confirmed the pathogenic role of HERV-W/MSRV and pointed out a double therapeutic effect of temelimab, which confers neuroprotective properties. On the one hand, temelimab prevents the release of proinflammatory cytokines induced by pHERV-W ENV in the blood and in the CNS and the consequent damage to the myelin sheaths. On the other hand, it allows OPC differentiation into oligodendrocytes to remyelinate demyelinated axons, preventing neuronal death. These properties make temelimab a potential targeted treatment for those neurological diseases in which HERV-W/MSRV causes inflammation and neurodegeneration, such as MS.

After the promising preclinical results, temelimab has been assessed in clinical trials as a new MS treatment under development. In a randomized, double-blind, placebo-controlled, dose escalation phase I trial, the safety of temelimab was assessed in a total of 88 healthy male volunteers with the administration of different doses ranging from 0.0025 to 6 mg/kg (ClinicalTrials.gov Identifier: NCT01699555 (accessed on 21 May 2021)) [218], 6 to 36 mg/Kg (ClinicalTrials.gov Identifier: NCT02452996 (accessed on 21 May 2021)) [221], and 36 to 110 mg/Kg (ClinicalTrials.gov Identifier: NCT03574428 (accessed on 21 May 2021)), in the three groups tested. All intravenously administered doses were well-tolerated, showing nonspecific minor adverse events. Its mean half-life was estimated to be between 19 and 28 days, depending on the dose, in line with the average of 3 weeks for IgG4 antibodies, so its application should follow a monthly regular basis [218,221]. In a phase IIa trial, the safety and pharmacokinetics of temelimab were assessed again but, this time, in 10 MS patients with monthly repeated doses of 2 mg/kg or 6 mg/Kg for 6 months (ClinicalTrials.gov Identifier: NCT01639300 (accessed on 21 May 2021)) [219,222]. Of those MS patients, one had relapsing-remitting MS (RRMS), three had primary progressive MS (PPMS) and six had secondary progressive MS (SPMS). Doses of 2 and 6 mg/Kg were selected to ensure a sustained pHERV-W ENV neutralization over 4 weeks. As in previous studies, temelimab was well-tolerated, its pharmacokinetics were dose-linear, and its mean half-life was between 27 and 37 days.

Surprisingly, it was observed that the administration of temelimab also decreased the genomic expression of HERV-W/MSRV *env* and *pol* mRNAs on PBMCs after three and six months of treatment. This was unexpected, as temelimab is a mAb designed to bind pHERV-W ENV, not its mRNA nor other HERV-W genomic sequences or features [222]. However, this effect on *env* mRNA has also been reported in other MS treatments, such as interferon β (IFN-β) [111,174] and the monoclonal antibody natalizumab [223]. Thus, importantly, temelimab, unlike other MS treatments, has been shown not to impair the immune system [224].

Later phase IIb (CHANGE-MS) (ClinicalTrials.gov Identifier: NCT02782858 (accessed on 21 May 2021)) [225] and phase IIc (ANGEL-MS) (ClinicalTrials.gov Identifier: NCT03239860 (accessed on 21 May 2021)) studies tested temelimab´s efficacy on RRMS in the long term [226]. The results obtained were encouraging, as neuroprotective and regenerative effects of temelimab were observed in RRMS patients, indicating that a new RRMS treatment with neuroprotective and antiretroviral effects, which does not impair the immune system, may be available in the close future. Furthermore, these results open the door to future clinical studies testing temelimab in progressive MS characterized by established neurodegeneration, and testing its efficacy on other HERV-W/MSRV-related diseases, such as CIDP, where HERV-W/MSRV triggers the innate immunity response by engaging the TLR4 receptor in HSCs, and releasing IL-6 and CXCL10 [190], as mentioned earlier. In fact, in vitro experiments have demonstrated a neutralizing effect of temelimab on the HERV-W/MSRV-mediated induction of HSCs, inhibiting the release of IL-6 and CXCL10 [190]. A phase II clinical trial is now ongoing in progressive MS with escalating doses of 18, 36 and 54 mg/Kg versus the placebo (ClinicalTrials.gov Identifier: NCT04480307 (accessed on 21 May 2021)).

Furthermore, HERV-W/MSRV not only has shown an impact on neurological diseases but, also, in other autoimmune diseases, such as type 1 diabetes (T1D), where pHERV-W ENV has been detected in the serum, PBCMs and pancreatic acinar cells of patients [227]. Interestingly, as previously mentioned for MS and schizophrenia, influenza and EBV also associate with T1D, as well as enterotropic Coxsackie B virus infections that transactivate HERV-W [228,229]. In vitro experiments suggest that HERV-W/MSRV triggers an inflammatory response and, also, directly affects insulin production by pancreatic β-cells [227]. HERV-W/MSRV has been previously shown to induce potent inflammatory effects by engaging with TLR4s [25,167], and pancreatic β-cells also express this receptor type [230]. Further, transgenic mice expressing HERV-W/MSRV display T1D clinical symptoms, such as hyperglycemia and hypoinsulinemia, and show a strong increase in the release of IL-6 and TNF-α into the bloodstream [227,231]. These results evidence a role for HERV-W in T1D pathogenesis.

When temelimab was assayed by in vivo and in vitro models of T1D, both the release of proinflammatory cytokines and the toxic effect of HERV-W/MSRV on pancreatic β-cells were inhibited [232]. These results allowed to carry out a double-blind, placebo-controlled Phase IIa clinical trial, RAINBOW-T1D (clinicaltrials.gov no NCT03179423 (accessed on 21 May 2021)), to assess the safety of temelimab on T1D patients and its effect on the autoimmune process. Temelimab was administered at a dose of 6 mg/kg in 60 patients for a year on a monthly regular basis [232,233]. As with MS patients, temelimab was well-tolerated in T1D patients. Furthermore, it reduced the frequency of hypoglycemia events and the levels of anti-insulin autoantibodies after six months of treatment.

In summary, temelimab represents the first HERV-directed treatment that also does not impair the immune system [224], unlike other MS treatments, especially for RRMS, which is the most common form of MS. As RRMS is mainly driven by inflammation, most treatments act on immune system cells trying to reduce the immune responses and inflammation. For example, natalizumab prevents the BBB crossing of lymphocytes, or rituximab depletes mature B-cell pools; even fingolimod reduces the number of circulating lymphocytes by preventing them from leaving secondary lymphoid organs [234]. Thus, most potential treatments have several safety concerns that have even led to their withdrawal, as it was the case of daclizumab. In this sense, temelimab shows an improved performance, with minimal adverse events in the so far completed clinical trials. Furthermore, unlike other MS treatments, temelimab shows neuroprotective effects in MS patients due to the neutralization of the pHERV-W ENV-induced blockade of OPC differentiation into oligodendrocytes. Beyond MS, temelimab seems to be effective in other HERV-W/MSRV-related diseases, such as CIDP or T1D. Therefore, it might be of interest to assess its efficacy in the treatment of additional diseases, such as fibromyalgia or chronic fatigue syndrome, where the upregulated expression of HERV-W has been recently reported [44,46].

Lastly, the GeNeuro company is presently developing an anti-HERV-K Env mAb for the treatment of ALS (patent number: US20200308258) [235]. Its application in vitro shows neuroprotective effects like those described for temelimab. It restores the activity of motor neurons and increases their viability and neurite length as compared to the untreated controls [235]. Thus, temelimab seems to represent the beginning of a promising new generation of antibody-based drugs directed specifically against HERV retroviral proteins with pathogenic properties.

## Figures and Tables

**Figure 2 pharmaceuticals-14-00495-f002:**
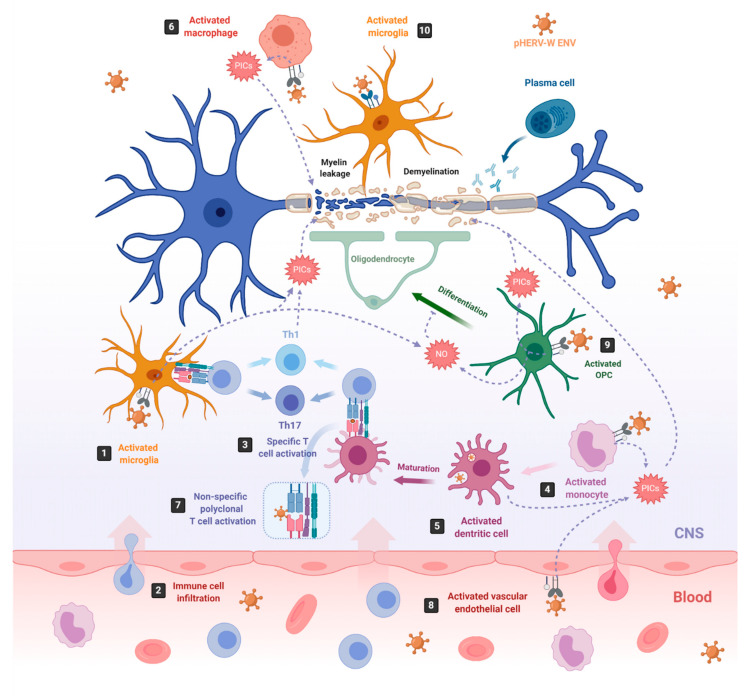
pHERV-W-ENV drives inflammation and neurodegeneration in MS. In MS, inflammation starts in the CNS with the activation of microglia, which release proinflammatory cytokines (PICs) that directly damage the myelin sheaths (**1**). Cell debris and PICs attract peripheral immune cells to the CNS (**2**); among them are T cells, which are then presented by APCs like microglia or infiltrating monocyte-derived dendritic cells (**3**) [152]. In this sense, pHERV-W ENV stimulates the innate immune system by engaging TLR4/CD14 receptors in microglia (**1**) monocytes (**4**), dendritic cells (**5**) and macrophages (**6**), inducing the release of PICs. In addition, it stimulates specific T-cell activation (**3**) and abnormal polyclonal T-cell activation (**7**) [24,25], acting as a superantigen. Furthermore, pHERV-W ENV activates vascular endothelial cells (**8**), making the migration of circulating cells possible at the level of ongoing brain lesions initiated by perivascular macrophages and/or microglia [164]. On the other hand, pHERV-WENV impairs the remyelination process by inducing the release of PICs and nitrosative stress in oligodendrocyte precursor cells (OPCs) (**9**) and the microglia (**1**), inhibiting the differentiation of OPCs to oligodendrocytes and damaging the myelin sheath, ultimately leading to neuronal loss [165]. pHERV-W ENV also drives microglial cells to physically interact with myelinated axons and induces the leakage of intra-axonal and myelin proteins (**10**) [112]. Created with Biorender.com (accessed on 21 May 2021).

**Table 1 pharmaceuticals-14-00495-t001:** HERVs associated with neurological diseases.

Disease	Retrotransposon	Elevated Cytokines	Ref.
**Multiple Sclerosis**	HERV-W ENV, *POL* HERV-H ENV	IFN-γ, IL-6, TNF-α, IL-17, IL-22, IL-12, IL-1	[26,27,28,29]
**Schizophrenia**	HERV-W ENV, GAG	IL-6, IL-1β, IL-2, sIL-2R, IL-8, IL-1, IL-1RA, TNF-α, IFN-γ, IL-4, TGF-β, IL-18, IL-10	[30,31,32,33,34]
**Bipolar disorder (BD)**	HERV-W *ENV*	IL-6, TNF-α, IL-10, sIL-2R, IL-1β, IL-1RA	[34,35,36]
**Autism spectrum disorder (ASD)**	HERV-H *ENV* HERV-K *ENV* HERV-W *ENV*	IFN-γ, IL-1β, IL-6, TNF-α, IL-10	[37]
**Attention deficit hyperactivity disorder (ADHD)**	HERV-H *ENV*	IL-6, TNF-β, IFN-γ, IL-2, IL-10, IL-13, IL-16	[38,39,40]
**Sporadic amyotrophic lateral sclerosis (sALS)**	HERV-K *GAG, POL, ENV*	TNF-α, IFN-γ, IL-6, IL-8, IL-1β, IL1RA, IL-2, IL-4, IL-5, IL-7, IL-9, IL-10, IL-12p70, IL-13, IL-15, IL-17, IL-18, IL-21	[41,42]
**Fibromyalgia (FM)**	HERV-W *POL* HERV-K *ENV* HERV-H *GAG*	INF-β, INF-γ, IL-1RA, IL-6, IL-17A,	[43,44]
**Chronic fatigue syndrome/** **myalgic encephalomyelitis (CFS/MS)**	HERV-K *POL*	IFN-γ, IFN-α, IL-4, IL-5, IL-7, IL-13, IL-6, IL-1β, IL-2, IL-12	[45,46]

## Data Availability

Not applicable.

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
