# Peer review of "Human Endogenous Retrovirus as Therapeutic Targets in Neurologic Disease"

_pharmaceuticals, 2021, doi:10.3390/ph14060495_

Round 1

Reviewer 1 Report

The authors in manuscript entitled “Human Endogenous Retrovirus as Therapeutic Targets in Neurologic Disease” have described the brief overview about the recent advances on the epigenetic mechanisms controlling HERV expression and the pathogenic effects triggered by HERV derepression

Authors have tried to convey the information that will be helpful in the design of new strategies to unveil epigenetic failures behind HERV-triggered disease, opening new possibilities for druggable targets and/or for extending the use of temelimab to treat other associated diseases.

To conclude, this review is a nice compact description about the use of HERV as therapeutic targets in neurological disease. The quality of the review is suitable for publication.

There are some minor issues which should be addressed:

1)            In a part of the introduction, it should be crisp and brief about the focused study.

2)            There are few typos and English and grammar errors which should be rectify.

Author Response

The authors in manuscript entitled “Human Endogenous Retrovirus as Therapeutic Targets in Neurologic Disease” have described the brief overview about the recent advances on the epigenetic mechanisms controlling HERV expression and the pathogenic effects triggered by HERV derepression

Authors have tried to convey the information that will be helpful in the design of new strategies to unveil epigenetic failures behind HERV-triggered disease, opening new possibilities for druggable targets and/or for extending the use of temelimab to treat other associated diseases.

To conclude, this review is a nice compact description about the use of HERV as therapeutic targets in neurological disease. The quality of the review is suitable for publication.

There are some minor issues which should be addressed:

  • In a part of the introduction, it should be crisp and brief about the focused study.

RESPONSE: Thank you. Part of the introduction has been removed or restructured to improve focusing and briefness.

  • There are few typos and English and grammar errors which should be rectify.

RESPONSE: The whole document has been proofread. Typos and grammar errors should have been corrected.

Reviewer 2 Report

The content of this review is abundant, but I hope that the list of candidate of targeting HERVs in various neurological diseases in a table. 

Author Response

The content of this review is abundant, but I hope that the list of candidate of targeting HERVs in various neurological diseases in a table. 

RESPONSE: Thank you. Please see the newly incorporated Table 1 which summarizes correlations established by the literature between HERV families and neurologic disorders.

Reviewer 3 Report

Overall the review by Giménez-Orenga and Oltra focusing on “Human Endogenous Retrovirus as Therapeutic Targets in Neurologic Disease” is well written and includes several interesting aspects. However, some more aspects need to be addressed before publication:

Major Comments:

  • There are several aspects that need to be addressed in the context of epigenetic control. For example the recent very interesting finding that m6 A RNA methylation can regulate the fate of endogenous retroviruses, which was recently published (PMID: 33442060), is not mentioned. Also another aspect that is really interesting and important in respect to epigenetic control of HERVs especially in the brain, is the fact that this control might be cell type specific due to different KRAB-ZNF proteins expressed in different cell types as shown recently by the group of Didier Trono (PMID: 32923624).
  • Several viruses are mentioned that can reactivate HERV expression. One virus that has not been mentioned is HIV although it is very well described that HIV-1 results in activation of multiple HERV families as described in for example in: PMID: 25056891 and PMID: This aspect should be included, especially as the authors later on write about TDP43 in connection to HIV-1 and HERVs.
  • As the Review is focusing on HERVs in Neurological Diseases another important aspect is the effect of HERVs during neuronal development, brain development and also inflammatory response. Several recent papers have shown that HERVs contribute to brain development and can influence inflammatory response. These aspects and studies need to be included. PMID: 33644903, PMID: 32271161, PMID: 32923624, PMID: 32669437 and Padmanabhan Nair et al. Cell Stem Cell 2021: DOI:https://doi.org/10.1016/j.stem.2021.04.009

Minor Comments:

  • Overall literature citations used are rather old. It would be good to include more recent publications. Especially as the field on HERVs and Neurological Diseases has gained more and more interest during the last 3 years.
  • Line 399: At the end of the sentence the point is missing.
  • Line 440: “HERV-K related diseases.“ The sentence is not complete or is this a heading?
  • Line 525: “…was observed that administration...“. One space between that and administration needs to be deleted.

Author Response

Overall the review by Giménez-Orenga and Oltra focusing on “Human Endogenous Retrovirus as Therapeutic Targets in Neurologic Disease” is well written and includes several interesting aspects. However, some more aspects need to be addressed before publication:

 Major Comments:

  • There are several aspects that need to be addressed in the context of epigenetic control. For example the recent very interesting finding that m6A RNA methylation can regulate the fate of endogenous retroviruses, which was recently published (PMID: 33442060), is not mentioned.

RESPONSE: Thank you. The corresponding reference (PMID: 33442060) has been added in lines 189-192, as reference number 92. for a more complete explanation of the regulatory mechanisms controlling HERV expression. The text now reads as follows: Another post-transcriptional regulatory mechanism is m6A RNA methylation, which reduces the half-life of ERV mRNAs, protecting cellular integrity, as shown in mouse cells [92]”.

  • Also another aspect that is really interesting and important in respect to epigenetic control of HERVs especially in the brain, is the fact that this control might be cell type specific due to different KRAB-ZNF proteins expressed in different cell types as shown recently by the group of Didier Trono (PMID: 32923624).

RESPONSE: Thank you. Lines 76-82 have been rephrased to include the indicated reference (PMID: 32923624) (reference number 48).

The text now reads as follows: “While most HERVs are constitutively repressed in the human genome, some co-opted elements must become activated in a limited way, at specific time frames, and in particular cell types [20,48]. To achieve this complex context-specific regulation, some HERVs harbor binding sequences for KRAB-containing zinc finger proteins (KRAB-ZFPs)”.

  • Several viruses are mentioned that can reactivate HERV expression. One virus that has not been mentioned is HIV although it is very well described that HIV-1 results in activation of multiple HERV families as described in for example in: PMID: 25056891 and PMID: This aspect should be included, especially as the authors later on write about TDP43 in connection to HIV-1 and HERVs.

RESPONSE: Thank you.  Reference PMID: 25056891 has been included in lines 523-528, as reference number 209, to highlight the relationship between HERV-K RNA and HIV disease.

The text now reads as follows: “TDP-43 was initially identified as a protein able to bind TAR DNA sequence motifs in the human immunodeficiency virus (HIV) [207]. Later, it was found to be located in the nucleus of neurons and glial cells, such as oligodendrocytes [206,208], however, its physiological role in the brain remains unknown. Besides HIV, another retroviral element which contains TDP-43 binding sites is the endogenous HERV-K (HML-2). Interestingly, HERV-K (HML-2) RNA is upregulated in PBMCs of HIV patients [209]”.

  • As the Review is focusing on HERVs in Neurological Diseases another important aspect is the effect of HERVs during neuronal development, brain development and also inflammatory response. Several recent papers have shown that HERVs contribute to brain development and can influence inflammatory response. These aspects and studies need to be included.

PMID: 33644903:

RESPONSE: The indicated reference has been included in lines 104-107, as the number 62, to improve the understanding of the role that KAP1 plays in ERV silencing, and to emphasize its particular importance during neuronal differentiation.

The text now read as follows:
Furthermore, KAP1 mediated silencing is especially important for brain development. Its depletion in neural progenitor cells aberrantly activates ERV expression while having no impact on adult neural cells [62]”

PMID: 32271161 (added as reference 203)
PMID: 32669437 (added as reference 199)
Padmanabhan Nair et al. Cell Stem Cell 2021 (added as reference 200)

RESPONSE: The corresponding references have been included in lines 542-558. The added references allow for an improved understanding of HERV-K´s role in neuronal differentiation and hence, a possible signaling pathway implicated in its neurodegenerative effects in ALS. Furthermore, HERV-K neurotoxicity in Alzheimer’s disease mediated by TLR8 signaling has also been explained, giving a broader view of HERV-K implication in neurodegenerative diseases.

The text now reads as follows:

“Additionally, HERV-K (HML-2) Env downregulation is necessary for differentiation of neural stem cells [212]. In this regard, transcriptional activation of the HERV-K (HML-2) group of elements impairs cortical neuronal differentiation, reducing neurite length and complexity. This effect was reported to be mediated by upregulation of the Neurotrophic Receptor Tyrosine Kinase 3 (NTRK3) gene [213], an important factor for the development of cortical neurons whose knockdown completely rescued this phenotype [213,214].

HERV-K (HML-2) is also involved in Alzheimer’s disease, an illness characterized by the accumulation of beta-amyloid protein aggregates (plaques) in the extracellular matrix and Tau protein aggregates (tangles) inside neurons. Progressive accumulation of these molecules causes synaptic loss and neurodegeneration [215]. HERV-K (HML-2) transcripts are found upregulated in CSF of Alzheimer’s patients in correlation with TLR8 RNA. HERV-K -induced neurotoxicity and neuronal apoptosis is mediated by TLR8 and SARM1 (Sterile alpha and TIR domain-containing 1) signaling. Furthermore, HERV-K RNA stimulates microglia release of proinflammatory molecules. However, microglia activation does not seem to be required for triggering neurotoxicity [216]”.

Reference 201 (PMID: 18003743), related to NTRK3 gene, and reference 202 (PMID: 31753135), related to Alzheimer’s disease, were added to support the information where needed.

  • PMID: 32923624.

RESPONSE: This reference was already added in lines 76-82 as previously mentioned.

Minor Comments:

  • Overall literature citations used are rather old. It would be good to include more recent publications. Especially as the field on HERVs and Neurological Diseases has gained more and more interest during the last 3 years.

RESPONSE: New publications on the role of HERV-K have been added (PMID: 32271161, PMID: 32669437).  

  • Line 399: At the end of the sentence the point is missing.

RESPONSE: The point has been added to the end of the sentence.

  • Line 440: “HERV-K related diseases.“ The sentence is not complete or is this a heading?

RESPONSE: It is a heading. The format has been corrected.

  • Line 525: “…was observed that administration...“. One space between that and administration needs to be deleted.

RESPONSE: This has been corrected.

Author Response

RESPONSE: No comments to be responded were received.

Round 2

Reviewer 3 Report

All concerns have been addressed and review has improved. Accept for publication :).